# Abdominal Organ Segmentation Method Using Attention-Enhanced nnUNet

Yuhua Xie[1], Shouyu Jiang[1], Liangduan Wu[1], and Kaifeng Zhou[1]

SiChuan University, Chengdu 610065, China
`xieyuhua@stu.scu.edu.cn`

**Abstract.** Medical image segmentation plays a pivotal role in clinical diagnosis and treatment. In particular, automatic segmentation of abdominal organs is crucial for computer-aided diagnosis, surgical navigation, and various medical applications. This article introduces an improved approach based on nnU-Net, a state-of-the-art neural network model for biomedical segmentation, with the integration of attention mechanisms. The proposed method enhances segmentation accuracy by introducing attention modules while preserving nnU-Net's established workflow. Three key preprocessing steps—image cropping, resampling, and normalization—are detailed to prepare abdominal CT images. Additionally, SE (Squeeze-and-Excitation) attention modules are incorporated into nnU-Net to improve feature representation and semantic segmentation accuracy. Experimental results on the FLARE 2023 dataset demonstrate the effectiveness of the proposed method, achieving a mean Dice Similarity Coefficient of 0.411, even with limited training data.

**Keywords:** nnU-Net · SE attention · Medical image segmentation.

## 1 Introduction

Medical image segmentation is playing an increasingly pivotal role in the realm of clinical diagnosis and treatment. Particularly, the automatic segmentation of abdominal organs stands as a crucial component within computer-aided diagnosis, surgical navigation, visual enhancement, radiotherapy, and biomarker measurement systems. The precise delineation of various organs and structures is paramount for guiding physicians in their diagnostic and treatment decisions. The MICCAI FLARE 2023 challenge encompasses the multifaceted task of multi-organ segmentation in abdominal CT scans, featuring a substantial dataset of 2200 partially labeled CT scans alongside an additional 1800 unlabeled CT scans. While nnunet has demonstrated commendable performance across diverse medical segmentation tasks, recent research articles have predominantly utilized nnunet as a baseline to underscore their own achievements in attaining state-of-the-art (SOTA) results, even if some of these state-of-the-art (SOTA) methods may not outperform nnunet on certain datasets. This article opts to preserve nnunet's intricately designed workflow while strategically introducing attention mechanisms exclusively in the model section. This approach aims to enhance

nnunet's overall segmentation performance in multi-organ segmentation tasks. The main contribution of this work are summarized as follows: ·Added attention module to nnunet to achieve higher segmentation accuracy

## 2   Method

### 2.1   Preprocessing

In this paper, we performed three operations on abdominal CT images: cropping, resampling and normalization. We will now provide a detailed introduction of these three operations.

**Image Cropping** The process of image cropping involves removing the region of the 3D medical image that is equal to zero. The specific method is to find the minimum 3D bounding box in the image, and set all values outside of this bounding box to zero, then use this bounding box to crop the image. Compared to before cropping, the cropped image has no impact on the final segmentation result, but it can reduce the size of the image, avoid useless computation, and improve computation efficiency. The implementation of image cropping can be divided into three steps:

1. Generate a 3D non-zero mask based on the image data, which indicates the regions in the image are non-zero.

2. Determine the bounding box to be used for cropping based on the generated non-zero template.

3. Crop the image based on the bounding box and then reassemble it. Modify the value of zero area in the mask to -1 for subsequent image processing operations.

**Image Resampling** The purpose of resampling is to solve the problem of inconsistent physical space sizes represented by voxels in different images in some 3D medical image datasets. This is because convolutional neural networks only operate on voxel space and ignore size information in actual physical space. To avoid such differences, it is necessary to change the size of the images to ensure that each voxel represented by different image data corresponds to the same actual physical space.

The specific values of image dimension changes are determined according to the expected target space. We calculate the spatial lengths of each image in each direction, take the median in each direction, and finally determine the expected target space based on the three medians. The specific resampling operation can also be divided into three steps.

1. According to the determined target spatial size for resampling and the presence of anisotropic issues, if one dimension is three times longer than the other two dimensions and the length of that dimension is less than one-third of the length of another dimension, then anisotropy exists. However, there is no anisotropy issue with the data here, so no extra operations are needed.

2. Determine the target size of each image based on the size of the target space.

3. Resize the image to the target size based on the computed dimensions.

**Image Normalization** Here, image normalization is performed using the mean and standard deviation of the entire training set, rather than calculating the mean and variance using only the grayscale information of a single image. There are two reasons for this. In the one hand, in CT images, the intensity information (HU value) can reflect the physical properties of different tissues. By using the statistical information of the entire training set, the additional information in the HU values can be effectively utilized. In the other hand, CT images often contain isolated outliers with extremely high or low values.

1. Collect statistical information on the foreground of the entire CT image training set. According to the processing steps in the image cropping, the value of -1 in the mask represents a background with a value of 0, and the value of 0 in the mask represents a non-zero background. To simplify the calculation, only 1/10 of the non-zero values are sampled for each image.

2. Calculate the mean, standard deviation, 0.5th percentile and 99.5th percentile of the entire training set, and use these statistical values to normalize each image.

**Image Resampling** The purpose of resampling is to solve the problem of inconsistent physical space sizes represented by voxels in different images in some 3D medical image datasets. This is because convolutional neural networks only operate on voxel space and ignore size information in actual physical space. To avoid such differences, it is necessary to change the size of the images to ensure that each voxel represented by different image data corresponds to the same actual physical space.

The specific values of image dimension changes are determined according to the expected target space. We calculate the spatial lengths of each image in each direction, take the median in each direction, and finally determine the expected target space based on the three medians. The specific resampling operation can also be divided into three steps.

1. According to the determined target spatial size for resampling and the presence of anisotropic issues, if one dimension is three times longer than the other two dimensions and the length of that dimension is less than one-third of the length of another dimension, then anisotropy exists. However, there is no anisotropy issue with the data here, so no extra operations are needed.

2. Determine the target size of each image based on the size of the target space.

3. Resize the image to the target size based on the computed dimensions.

**Image Normalization** Here, image normalization is performed using the mean and standard deviation of the entire training set, rather than calculating the

mean and variance using only the grayscale information of a single image. There are two reasons for this. In the one hand, in CT images, the intensity information (HU value) can reflect the physical properties of different tissues. By using the statistical information of the entire training set, the additional information in the HU values can be effectively utilized. In the other hand, CT images often contain isolated outliers with extremely high or low values.

1. Collect statistical information on the foreground of the entire CT image training set. According to the processing steps in the image cropping, the value of -1 in the mask represents a background with a value of 0, and the value of 0 in the mask represents a non-zero background. To simplify the calculation, only 1/10 of the non-zero values are sampled for each image.

2. Calculate the mean, standard deviation, 0.5th percentile and 99.5th percentile of the entire training set, and use these statistical values to normalize each image.

### 2.2    Proposed Method

In this project, we select nnU-Net as the foundational model. nnU-Net (no-new-Net) [2] is an outstanding neural network model for biomedical segmentation tasks, based on the U-Net network architecture. The core idea behind nnU-Net's design is to achieve excellent segmentation results across various datasets without manual parameter tuning, thanks to adaptive preprocessing and training strategies.

nnU-Net introduces U-Net cascades to address issues faced by 3D U-Net when segmenting large images. The first-level 3D U-Net is trained on downsampled images, and the results are upsampled to the original voxel resolution. These upsampled results, encoded as an additional input channel (one-hot encoding), are fed into the second-level 3D U-Net. Training is performed on full-resolution images using a block-based strategy [3]. The input image block size varies for different datasets, with '3D U-Net lowres' serving as the first level of the cascade, and the configuration of the second level aligning with that of the 3D U-Net network.

nnU-Net retains the U-Net network structure, focusing on data preprocessing. This allows it to automatically adapt to any dataset without manual intervention, effectively harnessing dataset characteristics for training the fundamental U-Net model.

### 2.3    Network Enhancement with SE Attention

In addition, this project enhances nnU-Net with the SE (Squeeze-and-Excitation) attention mechanism  [4]. The SE module aims to assign different weights to positions within an image from a channel perspective using a weight matrix. It accomplishes this by squeezing (Squeeze) and exciting (Excite) to obtain a 1x1xC weight matrix, which is used to reconstruct the original features, with different colors representing different values, quantifying channel importance.As shown in Figure 1.

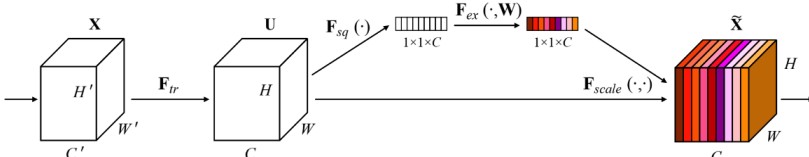

**Fig. 1.** SE Module Algorithm Flowchart.

The SE mechanism learns inter-channel relationships of feature maps, weighting different channel features to enhance the model's representation and performance. The primary applications of the SE attention mechanism are image classification and object detection, improving accuracy and robustness. In semantic segmentation tasks, SE attention can be applied to the encoder-decoder architecture to guide the model in capturing correlations between different semantic classes, thereby enhancing semantic segmentation accuracy and detail preservation. However, nnU-Net's integration with the SE attention mechanism hasn't been explored before, mainly due to:

nnU-Net primarily focuses on data preprocessing and achieves accuracy improvement with minimal changes to the U-Net framework, leading to the oversight of its potential.

SE attention mechanisms are often used in 2D image processing, and their effectiveness in enhancing 3D image segmentation accuracy hasn't been experimentally validated, introducing uncertainty.

SE attention modules can be added at different locations within the network, and their impact can vary. Inappropriate placement may lead to decreased training performance, posing risks. The large structure of the nnU-Net network increases the cost of trial and error.

Previous experiments indicate that applying the SE attention mechanism to U-Net can enhance its performance. By introducing the SE attention mechanism, U-Net can better learn relationships between features at different scales, improving semantic segmentation accuracy and detail preservation.

Therefore, this project boldly attempts to innovatively integrate SE attention modules into the nnU-Net network. Given the SE attention module's "plug-and-play" nature, it facilitates post-training modifications and multiple evaluation assessments.

The final placement of the SE module is as shown in Figure 2: within the generic UNet module's encoder-decoder structure, which occurs after the encoding phase and before the decoding phase. The U-Net encoder progressively reduces resolution to enlarge the receptive field, capturing more contextual information and extracting image features. The abstract features outputted by the encoder undergo squeezing and excitation operations by the SE attention module, generating a weight matrix for allocating weights to the original image. Finally, after passing through the U-Net decoder, which includes upsampling and

deconvolution operations, the spatial resolution of the feature maps gradually increases, resulting in segmentation predictions.

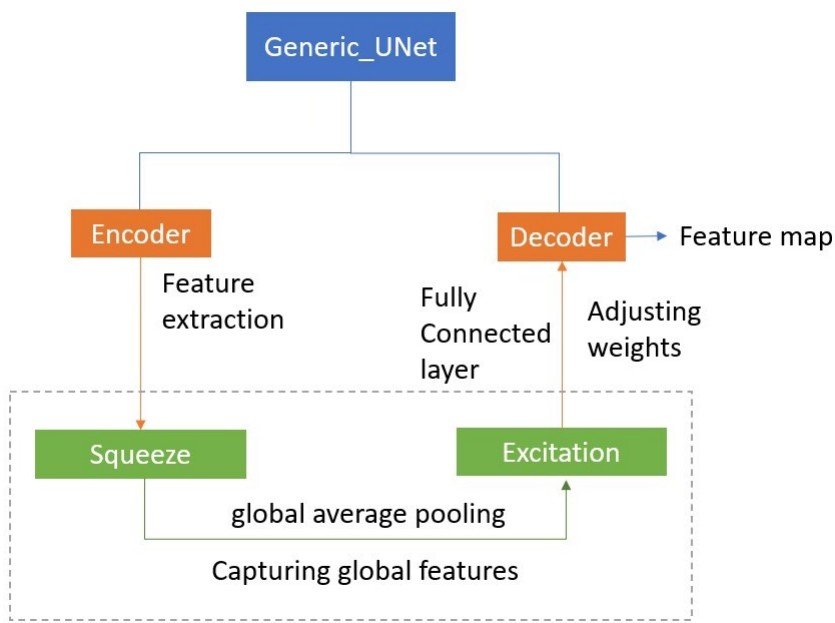

**Fig. 2.** SE Attention Module Integration Path and Principle.

### 2.4   Loss Function

The loss function used by the network is Lovász-Softmax Loss [1], which can be flexibly applied to multi-category segmentation. The core of Lovász-SoftmaxLoss is how many dimensions the image corresponds to. For example, a 5×5 image corresponds to a 25-dimensional loss function, instead of summing or averaging like other loss functions and then backpropagating. Through Optimizing the correct loss during training greatly improves the accuracy on semantic segmentation datasets. This loss function can greatly improve the segmentation quality, especially on small objects.

The network is trained using both Cross-Entropy Loss Function and Dice Loss Function in combination. The cross-entropy loss function can help the network quickly converge at the beginning of training, while the Dice loss function can help the network refine the target area.

Cross-Entropy Loss Function:

$$L_{CE} = -\frac{1}{N} \sum_{i=1}^{N} \sum_{c=1}^{M} y_{ic} \log(p_{ic})$$

where $N$ represents the total number of voxels, $M$ represents the number of categories, $y_{ic}$ represents the label value, and $p_{ic}$ represents the predicted value of the network.

$$L_{Dice} = 1 - \frac{2|P \cap Y|}{|P| + |Y|}$$

where $P$ represents the target area in the predicted results, while $Y$ represents the target area in the mask.

## 3   Experiments

The FLARE 2023 dataset includes 4000 3D CT scans, in which has 2200 cases with partial labels and 1800 cases unlabeled. Confronted with the enormity of the dataset, and constrained by limitations in our hardware environment, we chose only 200 cases for training, aims to assess the effectiveness of our improved methodology.

## 4   Results and discussion

### 4.1   Quantitative Results on Validation Set

Overall, as shown in Table 1, our method achieves a mean Dice Similarity Coefficient of 0.411.Figure 3 shows examples with segmentation results.

### 4.2   Limitations and Future Work

Our approach merely involved minor enhancements to the nnUNet model, without addressing issues such as lengthy inference and preprocessing times. During the course of this competition, we observed that traditional segmentation networks tend to exhibit shortcomings in segmentation results. Moreover, once a network is trained, the segmentation output remains the same for fixed inputs, and the model cannot efficiently rectify erroneous segmentation results at a lower cost. Therefore, following this competition, we have decided to introduce prompts into conventional segmentation models, hoping that interactive segmentation approaches can yield more favorable results.

## 5   Conclusion

This paper has presented an improved method based on nnUNet, augmenting its segmentation accuracy by introducing an attention mechanism while preserving nnUNet's original segmentation strengths. With only 200 cases as a training dataset, we achieved an average DSC coefficient of 0.411.

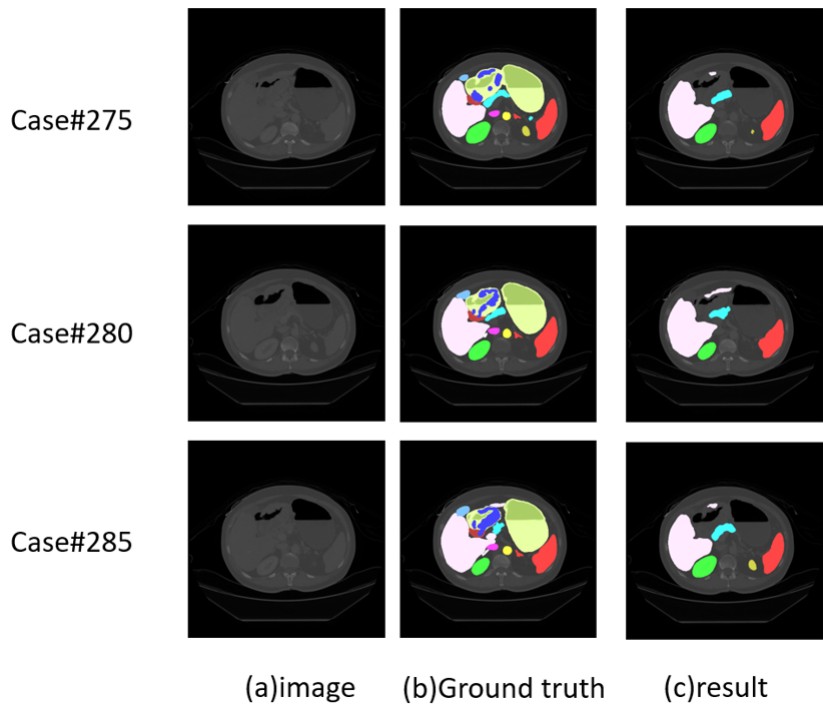

Case#275

Case#280

Case#285

(a)image        (b)Ground truth        (c)result

**Fig. 3.** SE Attention Module Integration Path and Principle.

**Table 1.** DSC values on different organs. Abbreviations: "Liv."-Liver, "RK"Right Kidney, "Spl."-Spleen, "Pan."-Pancreas, "Aor."-Aorta, "IVC"-Inferior Vena Cava, "RAG"-Right Adrenal Gland, "LAG"-Left Adrenal Gland, "Gall."-Gallbladder, "Eso."-Esophagus, "Sto."-Stomach, "Duo."-Duodenum, "LK"-Left Kidney, "Tu."-Tumor.

| metric | DSC |
|--------|-------|
| Liv. | 0.733 |
| RK | 0.640 |
| Spl | 0.757 |
| Pan | 0.587 |
| Aor | 0.446 |
| Ivc | 0.704 |
| RAG | 0.450 |
| LAG | 0.301 |
| Gall | 0 |
| Eso | 0.078 |
| Sto | 0.599 |
| Duo | 0.215 |
| LK | 0.696 |
| Tu | 0.489 |
| Avg | 0.411 |

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
