# OpenReview forum: "Abdominal Organ Segmentation Method Using Attention-Enhanced nnUNet"
_MICCAI.org/2023/FLARE — Submitted to FLARE 2023_

### Official Review · Reviewer_6a44 · 2023-09-21
**Abdominal Organ Segmentation Method Using Attention-Enhanced nnUNet**

**Rating:** 3
**Confidence:** 4

**Review:**

Pros:
 1. The proposed method enhances segmentation accuracy by introducing attention modules while preserving nnU-Net’s established
workflow with incorporating Squeeze-and-Excitation attention module to nnU-Net.

Cons：
 1. Please check MICCAI FLARE23 Template pdf, there is a number of things missing in your paper.

---

### Official Review · Reviewer_qkHW · 2023-09-27
**base on nnUNet  method  with attention module for abdominal organ segmentation**

**Rating:** 3
**Confidence:** 4

**Review:**

Pros:
The author introduce an SE(Squeeze-and-Excitation) module base on nnUnet to improve the segmentation results.



Cons:
1.You should check  that the method parts of your paper, such as image resampling, seem to be repeated.
2. the experients parts of the paper does not seem to be enough.

---

### Official Review · Reviewer_VeYT · 2023-10-02
**SE enhanced nnUnet submission**

**Rating:** 6
**Confidence:** 4

**Review:**

The paper describes the authors Flare23 challenge submission approach, with nnUnet  enhanced with the SE (Squeeze-and-Excitation) module. It  lacks details in Results (suggested by the paper template provided by the challenge organizers).  But since, this paper is only to accompany the challenge submission, it seems acceptable to miss some details in the results.

---

### Official Review · Reviewer_P9uX · 2023-10-21
**Abdominal Organ Segmentation Method Using Attention-Enhanced nnUNet**

**Rating:** 3
**Confidence:** 4

**Review:**

Pros:
- The paper introduces an SE (Squeeze-and-Excitation) attention module to nnU-Net, aiming to enhance the segmentation accuracy, which is a relevant addition to the field of medical image segmentation.
- The preprocessing steps (image cropping, resampling, and normalization) are well-documented and provide clarity on the data preparation process.

Cons:
- The paper seems to have repetitive sections in the method and image resampling parts, which may confuse readers. It is advisable to revise and streamline these sections for clarity and conciseness.
- The figures are not informative, and there are numerous formatting issues in text.

---

### Public Comment · ~PENGJU_LYU1 · 2023-11-26
**mandatory content missing**

please add results about  public,online validation, test quantitative resluts.
add efficienct measurement

---

### Decision · Program_Chairs · 2023-10-24

**Decision:**

Reject

**Comment:**

The authors didn't make responses to the valuable review comments.